# HPSERec: A Hierarchical Partitioning and Stepwise Enhancement Framework for Long-tailed Sequential Recommendation

**Xiaolong Xu**[1], **Xudong Zhao**[1], **Haolong Xiang**[1]*, **Xuyun Zhang**[2], **Wei Shen**[3], **Hongsheng Hu**[4], **Lianyong Qi**[5]

[1]School of Software, Nanjing University of Information Science and Technology, China
[2]School of computing, Macquarie University, Australia
[3]Independent Researcher, China
[4]School of Information and Physical Sciences, University of Newcastle, Australia
[5]College of Computer Science and Technology, China University of Petroleum (East China), China

## Abstract

The long-tail problem in sequential recommender systems stems from imbalanced interaction data, resulting in suboptimal model performance for tail users and items. Recent studies have leveraged head data to enhance tail data for diminish the impact of the long-tail problem. However, these methods often adopt ad-hoc strategies to distinguish between head and tail data, which fails to capture the underlying distributional characteristics and structural properties of each category. Moreover, due to a substantial representational gap exists between head and tail data, head-to-tail enhancement strategies are susceptible to negative transfer, often leading to a decline in overall model performance. To address these issues, we propose a hierarchical partitioning and stepwise enhancement framework, called HPSERec, for long-tailed sequential recommendation. HPSERec partitions the item set into subsets based on a data imbalance metric, assigning an expert network to each subset to capture user-specific local features. Subsequently, we apply knowledge distillation to progressively improve long-tail interest representation, followed by a Sinkhorn optimal transport-based feedback module, which aligns user representations across expert levels through a globally optimal and softly matched mapping. Extensive experiments on three real-world datasets demonstrate that HPSERec consistently outperforms all baseline methods. The implementation code is available at https://github.com/bolunxier123/HPSERec.

## 1 Introduction

Recommender systems have been widely adopted across various online platforms to suggest items that users are likely to engage with. Despite their remarkable success in numerous applications, the long-tail problem associated with both users and items remains a significant challenge, hindering the further development of recommender systems. On the user side, a small number of head users dominate interactions, far surpassing the engagement levels of the majority of tail users (1). As a result, models tend to focus disproportionately on head users, which ultimately reduces the platform's ability to retain new users. On the item side, most user interactions are concentrated on a limited number of popular items (2), leaving the majority of less popular items with minimal engagement. This concentration on popular items narrows the diversity of recommendations and fosters the emergence of filter bubbles (3; 4).

---

*Corresponding author. Email: hlxiang@nuist.edu.cn

39th Conference on Neural Information Processing Systems (NeurIPS 2025).

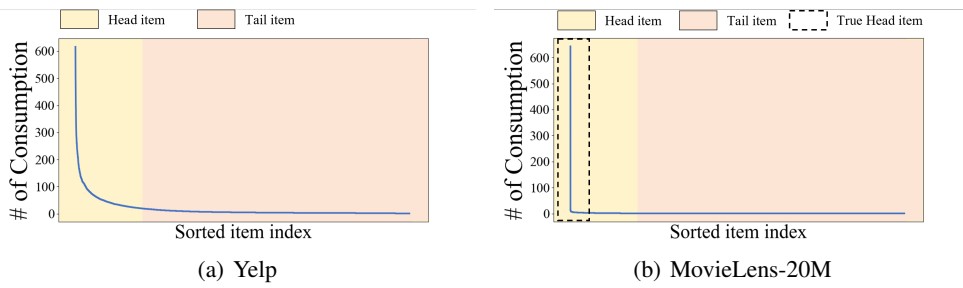

Figure 1: The number of consumed items on Yelp and MovieLens-20M

Recent studies focused on addressing the long-tail problem in sequential recommender systems (SRS). Several approaches address the long-tail problem from the user perspective, tackling the challenge of tail users by employing adversarial training to map head and tail users into a shared latent space, or by extending the interaction sequences of tail users (5; 6). Other methods focus on the item perspective, aiming to enhance the representation of tail items through attention mechanisms or by assigning higher weights to tail items during training (7; 8). In addition, certain works incorporate large language models (LLMs) to capture user sequence features and rich semantic information, enhancing the performance of long-tail data (9; 10; 11; 12).

Most existing methods follow a standard pipeline comprising data preprocessing, model training, and evaluation to deal with the long-tail problem, while those methods share two common limitations.

• **Arbitrary Partitioning**: During data preprocessing, most existing studies adopt a fixed-ratio approach to distinguish between head and tail data. For example, define the top 20% of items with the highest number of interactions as head items, and the remaining 80% as tail items. However, this ratio is largely based on empirical assumptions and may not generalize well across different application scenarios. As illustrated in Figure 1(a) and Figure 1(b), the item interaction distribution in the MovieLens-20M dataset exhibits a much stronger long-tail effect compared to the Amazon Yelp dataset. This variability suggests that fixed-ratio partitioning schemes may not generalize well across datasets, potentially leading to suboptimal modeling and recommendation performance.

• **Flawed Head-Tail Augmentation**: During model training, a common approach to mitigate the long-tail problem is to augment tail data using the rich information available in head data. For instance, information from head users is often transferred to tail users through shared or similar interaction sequences. However, due to the significant disparity in information density between head and tail entities, the presence of superficially similar patterns does not necessarily imply representational similarity. As a result, such augmentation may introduce noise rather than useful signal, ultimately degrading model performance.

To address the above problems, we propose a Hierarchical Partitioning and Stepwise Enhancement Framework for Long-tailed Sequential Recommendation (HPSERec), which leverages imbalance-aware partitioning, expert-based modeling, and distribution-level alignment to achieve balanced and effective recommendation. Our method comprises three modules: the distribution balance module, the feedforward module, and the feedback module. The distribution balance module addresses the problem of **Arbitrary Partitioning** by introducing a novel imbalance metric and a corresponding partitioning algorithm to ensure more balanced data subsets. The feedforward module tackles **Flawed Head-Tail Augmentation** by assigning expert networks to each subset and using progressive knowledge distillation to share tail item knowledge across the dataset. The feedback module further addresses this issue by refining upstream experts through soft distribution-level alignment using optimal transport, enabling adaptive updates without strict one-to-one matching.

Through this approach, HPSERec achieves more accurate recommendations for tail items without compromising recommendation performance for head items.

**Contributions.** The contributions of this study are summarized as follows:

• We define a metric to quantify data imbalance and design an algorithm to partition the item set based on this metric. The algorithm ensures that each subset minimizes data imbalance while maintaining a sufficient total number of interactions to preserve data utility.

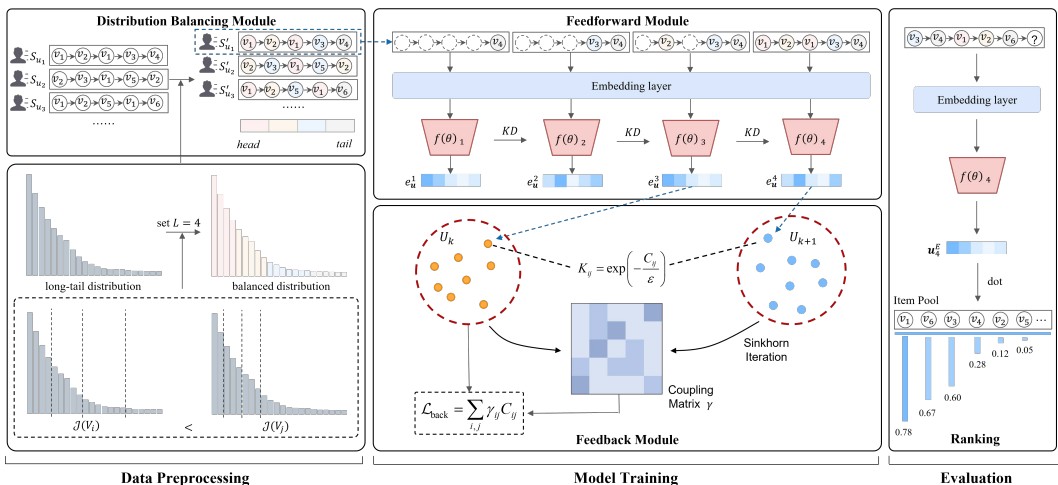

Figure 2: The overall architecture of HPSERec. Distribution balancing module splits the item set into subsets to reduce long-tail effects and improve training. Feedforward module boosts long-tail item representations with contrastive learning and refines global expert performance via knowledge distillation. Feedback module transfers head item information to long-tail experts using an annealing algorithm, expanding their receptive field.

- We propose a novel framework, called the Hierarchical Partitioning and Stepwise Enhancement Framework for Long-tailed Sequential Recommendation (HPSERec), which aims to address the long-tail problem in SRS.

- We are the first to address the long-tail problem in the recommendation domain by segmenting items and users into multiple subsets based on their long-tail characteristics, rather than merely classifying them into head and tail categories.

- We conduct extensive experiments on three real-world datasets, demonstrating that our approach improves the performance of tail items without compromising the performance of head items. Remarkably, its effectiveness surpasses even SRS integrated with LLM.

## 2 Problem Definition

Let $U$ and $V$ represent the sets of users and items. The historical interaction sequence of a user $u \in U$ is represented as:

$$S_u = [v_1, \ldots, v_k, \ldots, v_m], \tag{1}$$

where $m = |S_u|$ indicates the length of the sequence $S_u$, and $v_k$ represents the $k$-th item in the sequence. The embedding representations of a user $u$ and an item $v$ are denoted as $e_u \in \mathbb{R}^d$ and $e_v \in \mathbb{R}^d$, where $d$ represents the dimensionality of the hidden space. The goal of a SRS is to predict the next likely item $v_{|S_u|+1}$ based on the user's interaction sequence $S_u$ and a sequence encoder $f_\theta(\cdot)$. Formally, this process can be expressed as:

$$e_u^* = f_\theta(S_u) = \arg\max_{v_k \in \mathcal{V}} P(v_{m+1} = v_k | S_u), \tag{2}$$

where $e_u^*$ is representation of $u$. By performing a dot product between $e_u^*$ and all $e_v$, we obtain a ranked list $y = [y_1, y_2, \ldots, y_{|V|}]$, where $y_v$ represents the probability that $v$ is the next item $u$ will interact with. The top-$K$ items with the highest probabilities in $y$ are selected as the output of the recommender system.

# 3 Methodology

## 3.1 Overview

In this section, we introduce HPSERec, a framework designed to enhance traditional SRS by strengthening the representation of users' interests in long-tail items. HPSERec consists of three modules, including the distribution balancing module, the feedforward module, and the feedback module. The distribution balancing module (**Section 3.2**) partitions the set of items $V$ into multiple subsets with minimal long-tailed characteristics, ensuring that each subset can achieve optimal performance during subsequent training. The feedforward module (**Section 3.3**) enhances the embedding of long-tail items through contrastive learning-based data augmentation. This is followed by knowledge distillation, which transfers knowledge between expert networks to improve the long-tail performance of the global expert. The feedback module (**Section 3.4**) refines upstream experts via distribution-level alignment. It aligns current upstream user embeddings to a target distribution by minimizing the entropic regularized optimal transport cost using the Sinkhorn algorithm. Finally, we describe the overall training strategy of the framework (**Section 3.5**). The complete architecture of our model is illustrated in Figure 2.

## 3.2 Distribution Balancing Module

To more effectively address the intrinsic complexity of the long-tail phenomenon commonly observed in real-world recommendation datasets, we propose a novel and practical imbalance metric to precisely quantify the skewness in item interaction distributions.

The item set is denoted as $V = \{v_1, v_2, \ldots, v_n\}$, where each item $v_i$ is associated with an interaction count $c_i$. Assume that the items in $V$ are sorted in descending order according to their interaction frequencies. Our objective is to partition $V$ into $L$ disjoint item subsets $\{V_1, V_2, \ldots, V_L\}$, where each subset $V_k$ is defined as:

$$V_k = \{v_{T_{k-1}+1}, \ldots, v_{T_k}\}, \tag{3}$$

where $\{T_1, \ldots, T_{L-1}\}$ are index-based splitting points, with $T_0 = 0$ and $T_L = n$. For a given subset $V_k$, we define the normalized interaction probability distribution as:

$$p_i^{(k)} = \frac{c_i}{\sum_{j \in V_k} c_j}, \tag{4}$$

To quantify the concentration level of interactions within the subset, we introduce the following non-linear aggregation functional:

$$\Phi_k = \left( \sum_{i \in V_k} \left( p_i^{(k)} \right)^{\alpha} \right)^{\frac{1}{1-\alpha}}. \tag{5}$$

The value of $\Phi_k$ reflects the dispersion of item interactions within $V_k$. A smaller $\Phi_k$ indicates a more uniform distribution of interactions across items, while a larger $\Phi_k$ signifies that interactions are highly concentrated on a few dominant items. The parameter $\alpha$ is a tunable hyperparameter. When $\alpha < 1$, $\Phi_k$ is more sensitive to tail items, and when $\alpha > 1$, it becomes more responsive to head items.

To avoid extremely imbalanced partition sizes during the segmentation process, we further introduce a regularization term defined as:

$$B_k = \left( \frac{S_k - \mu}{\mu} \right)^2, \tag{6}$$

where $\mu = \frac{S_V}{L}$ denotes the expected average size of a partition, and $S_k$ is the size of subset $V_k$. This regularization term $B_k$ penalizes partitions that deviate significantly from the target size $\mu$, ensuring that each subset remains balanced not only in the interaction distribution but also in size.

Accordingly, for any given subset $V_k$, we define the overall imbalance score as:

$$\mathcal{J}(V_k) = \log \Phi_k + \gamma B_k, \tag{7}$$

where $\gamma \in [0, +\infty)$ is a hyperparameter that controls the strength of the regularization penalty. Thus, the optimal partitioning strategy can be formalized as the following discrete optimization problem:

$$\min_{\Theta} \quad \sum_{k=1}^{L} \mathcal{J}(V_k). \tag{8}$$

This formulation jointly controls interaction skewness and subset size, yielding balanced item groups for more effective head-tail modeling. The algorithm and proof details of the item subset partitioning process, based on our defined item interaction imbalance metric and implemented with dynamic programming and pruning, are provided in the Appendix A.1.

### 3.3 Feedforward Module

To enhance the performance of long-tail items, we initiate training by focusing on users' interests in long-tail items and gradually expand their preferences to encompass all items, while maintaining strong performance specifically on long-tail items. We design distinct expert networks to process corresponding datasets. For clarity, in the following sections, we refer to the expert responsible for inferring users' long-tail interests as the upstream expert and the expert responsible for inferring users' global interests as the downstream expert. Unlike traditional MOE models (13), where datasets assigned to each expert are statically partitioned, our approach progressively expands the datasets for each expert. For expert $E_i$, the user embedding $e_v^i$ represents the user's interest in tail items after excluding certain head items. Although $e_v^i$ does not fully capture the user's global preferences, it serves as a guiding signal for training subsequent experts.

We adopt a knowledge distillation approach (14), transferring knowledge from upstream experts to downstream experts. To mitigate performance degradation caused by substantial differences in the capacities of expert models, we restrict the knowledge distillation process to adjacent expert networks. This design ensures that downstream models can effectively learn from upstream models without being adversely affected by significant capability gaps.

The distillation loss is computed using a softened target distribution, where the smoothness of the logits is controlled by a temperature parameter $\tau$. The distillation loss is defined as:

$$\mathcal{L}_{KD} = \frac{1}{|\mathcal{V}_i|} \sum_{v \in \mathcal{V}_i} \mathrm{KL}\Big( \mathrm{Softmax}(z_v^{i-1}/\tau) \,\|\, \mathrm{Softmax}(z_v^i/\tau) \Big), \tag{9}$$

where $z_v^{i-1}$ and $z_v^i$ are the logits produced by experts $E_{i-1}$ and $E_i$, respectively.

### 3.4 Feedback Module

To enhance the representation quality of users in lower-level experts, we propose a Sinkhorn-based (15; 16) Feedback Module that adaptively transfers knowledge from higher-level experts via a principled optimal transport framework. This module aligns user representations across expert layers by computing a soft matching plan that minimizes the overall transport cost, thereby enabling fine-grained and globally optimal feedback supervision.

Let the user representation sets from two adjacent expert levels be denoted as $U^t = \{u_1^t, u_2^t, \dots, u_n^t\}$ and $U^{t+1} = \{u_1^{t+1}, u_2^{t+1}, \dots, u_n^{t+1}\}$, where each vector is normalized. We define the transport cost matrix $C \in \mathbb{R}^{n \times n}$ as:

$$C_{ij} = 1 - \cos(u_i^t, u_j^{t+1}), \tag{10}$$

which captures the pairwise dissimilarity between user representations across levels using cosine distance. Through this distance-based cost matrix, we formulate the entropy-regularized optimal transport problem as:

$$\min_{\gamma \in \Pi(\mu, \nu)} \sum_{i,j} \gamma_{ij} C_{ij} - \varepsilon \sum_{i,j} \gamma_{ij} \log \gamma_{ij}, \tag{11}$$

where $\gamma \in \mathbb{R}^{n \times n}$ is the transport plan, $\varepsilon > 0$ is a regularization parameter, and $\mu = \nu = \frac{1}{n}$ are uniform marginals. This formulation enables efficient, differentiable, and globally consistent feedback propagation via Sinkhorn iterations, facilitating stable optimization and improving representation alignment across expert layers in long-tailed scenarios.

The feedback supervision is imposed through the expected cost under the transport plan:

$$\mathcal{L}_{\text{back}} = \sum_{i,j} \gamma_{ij} C_{ij}. \tag{12}$$

This loss encourages each user representation in the current expert to softly align with structurally similar representations from the higher-level expert, thus enhancing representation consistency and improving learning in sparse regions of the long-tail distribution. Unlike hard matching schemes, our soft alignment mechanism, grounded in optimal transport theory, offers greater flexibility and robustness by leveraging its ability to model structured correspondences under distributional shifts, especially beneficial for handling noisy or sparse user signals. The detailed proof of this module is provided in the Appendix A.2.

### 3.5 Training Strategy

The HPSERec framework adopts a dual-stage training paradigm to leverage the interaction between the upstream and downstream expert models. In the forward stage, HPSERec emphasizes the use of upstream models to guide downstream models in discovering latent user features, particularly those associated with long-tail interests. In contrast, in the feedback stage, HPSERec aims to enhance user representations in upstream models by leveraging the global interest features extracted by downstream models. We have designed an alternating two-stage training strategy for the entire framework.

**Feedforward Stage.** During this stage, the training process starts with the top-level expert model and gradually incorporates adjacent downstream expert models. The downstream experts are treated as student networks, guided by the knowledge distilled from the upstream models. The overall loss function for this stage is defined as:

$$\mathcal{L}_{\text{forw}} = \mathcal{L}_{\text{rec}} + \beta \mathcal{L}_{KD}, \tag{13}$$

where $\mathcal{L}_{\text{rec}}$ denotes the recommendation loss, and $\mathcal{L}_{KD}$ corresponds to the knowledge distillation loss. The weight $\beta$ controls the weighting of the user's long-tail interest representation relative to the global interest.

**Feedback Stage.** In this stage, the training direction is reversed. This phase starts from the bottommost global expert and proceeds sequentially to the upstream adjacent experts. The goal is to address the knowledge gaps in the long-tail experts, improving their ability to capture users' long-tail interests. The loss function for this stage is $\mathcal{L}_{\text{back}}$, which can be calculated by Eq.(14). The detailed training procedure is outlined in Appendix A.3.

## 4 Experiment

### 4.1 Experimental Settings

**Datasets.** We compare HPSERec with baseline models using three real-world datasets from online services: Yelp, Amazon Beauty, and Amazon Music. For data pre-processing, we follow previous studies by excluding users with less than five interactions. More details about the datasets and preprocessing can be seen in Appendix B.1.

**Evaluation Metrics.** Performance is evaluated using top-k ranking metrics, specifically Hit Rate (HR@10) and Normalized Discounted Cumulative Gain (NDCG@10) (5; 17). Consistent with prior research SAS, we randomly select 100 items that the user has not engaged with to serve as negative samples (18; 19), paired with the ground truth, for metric computation.

**Compared Methods.** To assess the effectiveness of HPSERec, we compared it with two standard SRS models: SASRec (18) and BERT4Rec (17), two traditional enhancement framework for the long-tailed sequential recommendation: CITIES (7) and MELT (2), and three large model-based SRS approaches: RLMRec (10), LLMInit (12; 20) and LLM-ESR (21). The more details about baselines are put in Appendix B.2.

**Implementation Details.** To assess the effectiveness of each method, we initially identified the best hyperparameters with a single seed applied to the validation set. Subsequently, each method was trained with the selected hyperparameters across five different random seeds. More details about the datasets and preprocessing can be seen in Appendix B.3.

Table 1: The overall results of competing baselines and our HPSERec. The boldface refers to the highest score and the underline indicates the next best result of the models. "*" indicates the statistically significant improvements (*i.e.*, two-sided t-test with $p < 0.05$) over the best baseline.

| Dataset | Model | Overall | | Tail Item | | Head Item | | Tail User | | Head User | | Improv. |
|---|---|---|---|---|---|---|---|---|---|---|---|---|
| | | HR@10 | ND@10 | HR@10 | ND@10 | HR@10 | ND@10 | HR@10 | ND@10 | HR@10 | ND@10 | |
| Beauty | Bert4Rec | 0.3945 | 0.2453 | 0.0342 | 0.0085 | 0.4917 | 0.2922 | 0.3845 | 0.2384 | 0.4593 | 0.2941 | 33.9% |
| | SASRec | 0.4488 | 0.2861 | 0.1593 | 0.0856 | 0.7187 | 0.4815 | 0.4236 | 0.2690 | 0.5261 | 0.3611 | 17.7% |
| | CITIES | 0.3894 | 0.2462 | 0.1127 | 0.0013 | 0.4974 | 0.2745 | 0.3852 | 0.2249 | 0.4554 | 0.2594 | 35.6% |
| | MELT | 0.4869 | 0.3144 | 0.1598 | 0.0628 | **0.8055** | **0.5595** | 0.4719 | 0.3021 | 0.5523 | 0.3679 | 8.5% |
| | RLMRec | 0.4077 | 0.2565 | 0.1924 | 0.1660 | 0.6302 | 0.4657 | 0.4356 | 0.3016 | 0.4892 | 0.3345 | 29.5% |
| | LLMInit | 0.4351 | 0.2914 | 0.2714 | 0.1708 | 0.6984 | 0.5198 | 0.4919 | 0.3117 | 0.5430 | 0.3632 | 21.4% |
| | LLM-ESR | 0.4945 | 0.3275 | 0.2986 | 0.1713 | 0.7270 | 0.5232 | 0.4821 | 0.3103 | 0.5501 | 0.3425 | 6.8% |
| | HPSERec | **0.5281***| **0.3665*** | **0.3203*** | **0.2060*** | 0.7306 | 0.5229 | **0.5163*** | **0.3557*** | **0.5799*** | **0.4148*** | - |
| Yelp | Bert4Rec | 0.5307 | 0.3025 | 0.0131 | 0.0045 | 0.6834 | 0.3913 | 0.5319 | 0.3036 | 0.5251 | 0.2978 | 28.6% |
| | SASRec | 0.5866 | 0.3536 | 0.0890 | 0.0386 | 0.8002 | 0.4888 | 0.5848 | 0.3525 | 0.5945 | 0.3585 | 16.4% |
| | CITIES | 0.5745 | 0.3404 | 0.0776 | 0.0341 | 0.7648 | 0.4573 | 0.5751 | 0.3416 | 0.5891 | 0.3419 | 18.8% |
| | MELT | 0.6038 | 0.3687 | 0.0697 | 0.0263 | 0.8245 | 0.5041 | 0.6037 | 0.3688 | 0.6042 | 0.3681 | 13.1% |
| | RLMRec | 0.5306 | 0.3909 | 0.0104 | 0.0140 | 0.7683 | 0.4568 | 0.5351 | 0.3065 | 0.5137 | 0.2936 | 28.7% |
| | LLMInit | 0.6099 | 0.3781 | 0.0874 | 0.0330 | 0.7766 | 0.4797 | 0.6204 | 0.3795 | 0.6187 | 0.3823 | 11.9% |
| | LLM-ESR | 0.6190 | 0.3784 | 0.1584 | 0.0670 | 0.8045 | 0.5055 | 0.6138 | 0.3761 | 0.6331 | 0.3844 | 10.3% |
| | HPSERec | **0.6827*** | **0.4231*** | **0.3252*** | **0.1832*** | **0.8361*** | **0.5261*** | **0.6884*** | **0.4280*** | **0.6583*** | **0.4027*** | - |
| Music | Bert4Rec | 0.4721 | 0.3056 | 0.1222 | 0.0494 | 0.8299 | 0.5929 | 0.4475 | 0.2870 | 0.5638 | 0.3752 | 39.6% |
| | SASRec | 0.5031 | 0.3345 | 0.2243 | 0.0832 | 0.8328 | 0.6124 | 0.4835 | 0.3237 | 0.6317 | 0.4364 | 31.0% |
| | CITIES | 0.4421 | 0.2710 | 0.0824 | 0.0312 | 0.8347 | 0.5391 | 0.4192 | 0.2609 | 0.5082 | 0.3085 | 49.1% |
| | MELT | 0.5442 | 0.3832 | 0.3271 | 0.1539 | 0.8531 | 0.6292 | 0.5070 | 0.3374 | 0.6677 | 0.4722 | 21.1% |
| | RLMRec | 0.5431 | 0.3714 | 0.2473 | 0.1405 | 0.8511 | 0.6256 | 0.4946 | 0.3426 | 0.6604 | 0.4631 | 21.3% |
| | LLMInit | 0.5537 | 0.3877 | 0.3024 | 0.1574 | 0.8312 | 0.6426 | 0.5145 | 0.3591 | 0.6843 | 0.4746 | 19.1% |
| | LLM-ESR | 0.5958 | 0.4035 | 0.3318 | 0.1548 | 0.8961 | **0.6835** | 0.5672 | 0.3824 | 0.7069 | 0.4846 | 10.6% |
| | HPSERec | **0.6592*** | **0.4786*** | **0.4425*** | **0.2959*** | **0.8989*** | 0.6806 | **0.6428*** | **0.4701*** | **0.7144*** | **0.5069*** | - |

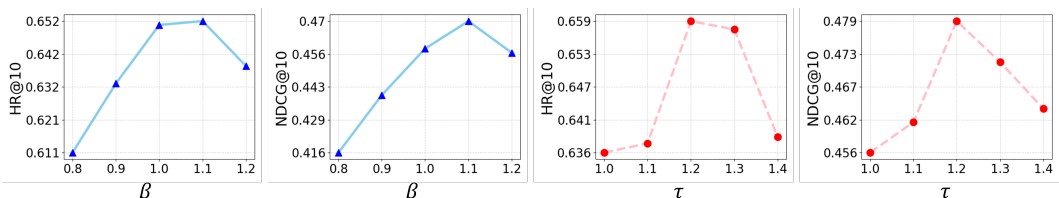

Figure 3: The hyper-parameter experiments on the weight of user's long-tail interest representation $\beta$ and distillation temperature $\tau$. The result are based on the Music dataset with the SASRec model.

## 4.2 Overall Performance Comparison

The comparative evaluation of HPSERec and other baseline models in three datasets is shown in Table 1, which includes overall recommendation performance along with specific performance metrics for head and tail users, as well as head and tail items. For the sake of comparison, we categorize the top 20% as head data and the remaining data as tail data.

Upon examining the table, we find that HPSERec significantly improves overall performance across all datasets. Specifically, in terms of HR@10, HPSERec outperforms the best baseline model, LLM-ESR, by 6.8% (Beauty), 10.3% (Yelp) and 10.6% (Music).

Compared to models specifically designed for the long-tail problem, HPSERec delivers substantial performance gains for both head and tail items, with especially notable improvements for tail items. Although HPSERec lacks a dedicated module for enhancing user representations, it still achieves optimal performance for both head and tail users. This suggests that enhancing both item and user performance is not always necessary. Since users and items share the same vector space, improving item performance also benefits user performance.

HPSERec significantly enhances the performance of tail items while maintaining strong performance for head items. In contrast, CITIES demonstrates a trade-off, as it often sacrifices the performance of head items across most datasets to boost the performance of tail items, creating seesaw problem. This stark contrast underscores the advantage of HPSERec, which effectively balances performance across both head and tail groups, addressing the long-tail challenge more comprehensively and equitably.

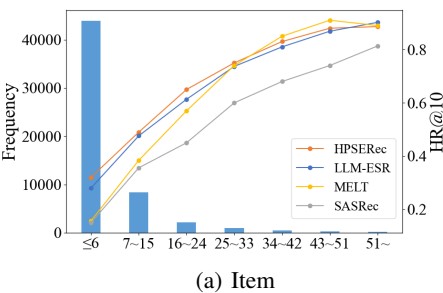 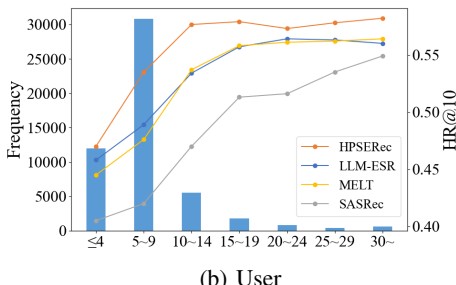

(a) Item              (b) User

Figure 4: The results of the proposed HPSERec and competing baselines in user and item groups. The results are based on the Beauty dataset with the SASRec model.

Table 2: The ablation study on the Amazon Music dataset with SASRec as the backbone SRS model. The boldface refers to the highest score and the underline indicates the next best result of the models. (DB: Distribution Balancing Module, FF: Feedforward Module, FB: Feedback Module)

| Dataset | Row | DB | FF | FB | Overall HR@10 | Overall ND@10 | Head User HR@10 | Head User ND@10 | Tail User HR@10 | Tail User ND@10 | Head Item HR@10 | Head Item ND@10 | Tail Item HR@10 | Tail Item ND@10 |
|---------|-----|----|----|----|------|------|------|------|------|------|------|------|------|------|
| Music | 1 | | | | 0.5431 | 0.3714 | 0.6843 | 0.4746 | 0.5149 | 0.3591 | 0.8511 | 0.6256 | 0.2473 | 0.1405 |
| | 2 | | ✓ | | 0.5889 | 0.3977 | 0.6661 | 0.4659 | 0.5401 | 0.3875 | 0.8655 | 0.6341 | 0.3461 | 0.1982 |
| | 3 | | ✓ | ✓ | 0.6094 | 0.4163 | 0.6729 | 0.4921 | 0.5803 | 0.4127 | 0.8730 | 0.6631 | 0.3640 | 0.2174 |
| | 4 | ✓ | ✓ | | 0.6122 | 0.4314 | 0.7020 | 0.5016 | 0.5726 | 0.4105 | 0.8532 | 0.6431 | 0.3782 | 0.2399 |
| | 5 | ✓ | ✓ | ✓ | **0.6524** | **0.4696** | **0.7212** | **0.5248** | **0.6175** | **0.4314** | **0.8943** | **0.6798** | **0.4154** | **0.2765** |

## 4.3 Hyper-parameter Analysis

To investigate the effects of the hyper-parameters in HPSERec, we show the performance trend along with their changes in Figure 3. The hyper-parameter $\beta$ controls the weighting of the user's long-tail interest representation relative to the global interest. With $\beta$ ranging from 0.8 to 1.2, the recommending accuracy rises first and drops then. A larger value of $\beta$ leads to suboptimal performance because it overemphasizes the user's long-tail representations, thereby compromising the recommendation accuracy for head items. Conversely, a smaller $\beta$ weakens the model's ability to capture long-tail item characteristics, ultimately degrading overall performance. As for the distillation temperature $\tau$, the optimal value is found to be 1.2. This is because a smaller $\tau$ limits the amount of informative signal conveyed by the upstream expert outputs, while a larger $\tau$ makes it more difficult for the downstream expert to effectively learn from the softened distributions. Additional experimental results can be found in Appendix C.1.

## 4.4 Group Analysis

To conduct a more detailed analysis of HPSERec's performance, we divided users and items into seven groups based on user sequence lengths and item popularity, the performance of each group is shown in Figure 4. From the results, we find that HPSERec consistently outperforms the baseline across all sequence lengths. This indicates that, although our model does not explicitly optimize for tail users, the enhancement of item representations also strengthens user embeddings. On the item side, we observe that MELT achieves optimal performance in the head group but underperforms in the extreme long-tail. In contrast, HPSERec demonstrates superior performance in the extreme long-tail while maintaining strong performance in the head group. Compared to the LLM-ESR, HPSERec achieves significant improvements, highlighting the advantage of our framework in enhancing the performance of tail items. Additional experimental results can be found in Appendix C.2.

## 4.5 Ablation Studies

The results of the ablation study are presented in Table 2. First, to evaluate the impact of the Distribution Balancing Module, we replaced it with a conventional classification scheme that separates head and tail classes based on item popularity, where the top 20% most interacted items are considered head items and the remaining are classified as tail items. A comparison between Rows 2 and 4, as well as Rows 3 and 5, demonstrates that our proposed method, which quantifies data imbalance

and enables knowledge transfer across subgroups, significantly improves overall performance. This improvement confirms the effectiveness of our subclass partitioning strategy, which is based on the imbalance characteristics defined earlier in the paper. Next, we examined the effect of the Feedback Module by removing it from two variants: one with the Distribution Balancing Module and one without, while keeping other components unchanged. Comparing Rows 2 and 3, as well as Rows 4 and 5, shows that the Feedback Module further enhances performance, working complementarily with the Feedforward Module. These results validate the design rationale of each module in HPSERec. Additional experimental results can be found in Appendix C.3.

## 5 Related work

**Sequential Recommendation.** The goal of sequential recommendation is to predict the next potential interaction item based on a user's historical interactions (22; 23; 24; 25; 26). These models are designed to capture the evolving preferences of users users over time. Traditional sequential recommendation systems often rely on Markov chain models (27), which excel at modeling user-item interactions. However, Markov models are limited to capturing short-term dependencies, making them less effective in real-world recommendation scenarios. Subsequent research has shifted toward neural networks as the primary approach for sequence modeling. GRU4Rec (28) employs recurrent neural networks to capture the sequential relationships in user-item interaction histories, enabling the prediction of the next potential item of interest. In contrast, SASRec (18) leverages attention mechanisms to model user sequences and capture global user interests. Bert4Rec (17) introduces the Cloze task to the sequential recommendation, training a bidirectional model to extend SASRec. TempRec (29) introduces a time-diversity-sensitive approach to news recommendation, using Transformer-based sequential modeling to capture temporal patterns effectively. However, these methods overlook the long-tail problem inherent in recommendation systems, resulting in poor performance on tail data.

**Long-tail Recommendation.** Despite significant progress in SRS, the long-tail problem remains underexplored. Regarding the issue of long-tail users, INSERT (5) formulates the recommendation task as a Few-Shot Learning problem, seeking similar sequences from other users and leveraging useful prior knowledge from different sessions to enhance recommendations for long-tail users. Similarly, ASREP (6) addresses this issue by generating pseudo-prior items through training with reversed sequences, effectively augmenting the sequence data for long-tail users. For long-tail items, CITIES (7) employs a self-attention mechanism to enhance the performance of tail items by leveraging head items, while Tail-Net (8) explicitly weights items within each user's sequence during inference based on the ratio of head to tail items. Among these works, MELT (2) is the only approach that simultaneously addresses both challenges, mitigating the issues of user and tail item by enabling mutual enhancement between user and item embeddings.

Due to their powerful reasoning and learning capabilities, LLMs have recently garnered significant attention in the field of long-tail recommendation (30; 31). Many researchers have explored leveraging LLMs to enhance the representations of long-tail items and users. RLMRec (10) utilizes LLMs to generate user and item profiles, effectively capturing their interaction preferences. LLMInit (12; 20) employs LLM-generated embeddings to initialize the embedding layer in SRS models. Additionally, LLM-ESR (21) adopts a dual-view modeling framework augmented with a retrieval-enhanced self-distillation method, improving the performance of long-tail users and items.

However, these methods overlook two critical aspects: (1) how to leverage the characteristics of the data to partition head and tail classes effectively, and (2) how to divide items into multiple subsets to reduce semantic disparities among them, thereby enabling more effective data augmentation.

## 6 Conclusion

In this work, we propose HPSERec, a novel framework to address the challenges posed by the long-tail problem in SRS. First, we define metrics to quantify the long-tail characteristics of the data and design a dynamic programming algorithm to partition the dataset, ensuring that the resulting subsets are as balanced as possible. HPSERec prioritizes learning the features of tail items and subsequently employs a combination of MoE and knowledge distillation techniques to transfer knowledge effectively. Following this, an annealing algorithm is applied to enhance the representation

of long-tail interests. This enhancement leverages users' global preferences, guided by the model's training progress and the similarity of interaction sequences between adjacent experts. Experimental results on three publicly available datasets demonstrate that HPSERec outperforms all baselines, including sequential recommendation models integrated with LLM. Furthermore, ablation studies confirm that HPSERec significantly enhances the performance of both tail items and tail users.

## Acknowledgments

This work was supported in part by the National Natural Science Foundation of China under Grant 92267104, and Jiangsu Provincial Major Project on Basic Research of Cutting-edge and Leading Technologies, under grant no. BK20232032.

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

# A Supplement to Method

In this section, the details of prompt design and the HPSERec procedures are addressed.

## A.1 Distribution Balancing Module

In Section 3.2, we introduced the formulation of data imbalance. In this section, we provide a detailed description of the algorithmic procedure used to partition the item set into balanced subsets.

Let $I(\Theta) = \sum_{k=1}^{L} \mathcal{J}(V_k)$, where $\Theta = \{T_1, \ldots, T_{L-1}\}$ is the set of threshold.

Algorithm 1 describes the dynamic programming approach used to identify the optimal threshold values $\Theta$ that minimize the overall imbalance score $I(\Theta)$.

---

**Algorithm 1:** Distribution Balancing Module

---

**Data:** An array $V$ storing the set of items, and an array $N$ storing the interaction counts for items in the set.
**Result:** The list of thresholds $\Theta$.

1  Sort $V$ in descending order based on item interaction counts;
2  Initialize $dp[0:|V|, 0:L] \leftarrow \infty$;
3  Initialize $path[0:|V|, 0:L] \leftarrow -1$;
4  **for** $l \leftarrow 1$ **to** $L$ **do**
5     **for** $i \leftarrow 1$ **to** $|V|$ **do**
6        **for** $j \leftarrow l - 1$ **to** $i$ **do**
7           Calculate $J(V[j:i])$ by Eq. (7);
8           **if** $H(V[j:i]) + dp[i][l-1] < dp[i][l]$ **then**
9              $dp[i][l] \leftarrow H(V[j:i]) + dp[i][l+1]$;
10             $\Theta[l-1] \leftarrow j$;
11          **end**
12       **end**
13    **end**
14 **end**
15 **return** $\Theta$;

---

Based on the threshold set $\Theta$, the item subsets $V_1', V_2', \ldots, V_L' \subseteq V$ are generated after partitioning. To better enable long-tail experts to guide the training of downstream experts, we propose that gradually expanding the range of input items for each expert, rather than providing disjoint item sets, is more effective in allowing long-tail experts to play an active role in subsequent training. The final subset assigned to each expert model is defined as:

$$V_k = \bigcup_{i=1}^{k} V_i' \tag{14}$$

where $V_k$ represents the item input range for the $k$-th expert $E_k$ in the feedforward module. When $k$ approaches 1, the input user sequence contains only tail items, causing the model to focus on capturing the user's tail-item preferences. Conversely, as $k$ approaches $L$, the input user sequence closely resembles the user's complete interaction history, prompting the model to capture the user's global interests.

## A.2 Feedback Module

In this section, we present a detailed proof of the feedback module. For clarity of exposition, we begin by redefining the following notations.

User embeddings produced by the $t$-th expert can be defined as:

$$U_t = \{e_1^t, \ldots, e_n^t\}, \tag{15}$$

while user embeddings produced by the $t + 1$-th expert can be defined as:

$$U_t = e_1^{t+1}, \ldots, e_n^{t+1}. \tag{16}$$

We define the cost matrix $C \in \mathbb{R}^{n \times n}$ using cosine distance:

$$C_{ij} = 1 - \frac{\langle u_i^t, u_j^{t+1} \rangle}{\|u_i^t\| \cdot \|u_j^{t+1}\|} \in [0, 2] \tag{17}$$

Since users are sampled uniformly during training, we assume that both user distributions are uniform:

$$\mu = \nu = \left( \frac{1}{n}, \ldots, \frac{1}{n} \right) \in \Delta_n \tag{18}$$

We consider the following entropy-regularized optimal transport objective:

$$\min_{\gamma \in \Pi(\mu, \nu)} \langle \gamma, C \rangle - \varepsilon H(\gamma), \tag{19}$$

where $\gamma \in \mathbb{R}_+^{n \times n}$ is the transport plan, $H(\gamma) = -\sum_{i,j} \gamma_{ij} \log \gamma_{ij}$ is the Shannon entropy, and $\Pi(\mu, \nu) = \{ \gamma \in \mathbb{R}_+^{n \times n} \mid \gamma \mathbf{1} = \mu, \ \gamma^\top \mathbf{1} = \nu \}$.

Let $f(\gamma) = \langle \gamma, C \rangle - \varepsilon H(\gamma)$.

**Lemma 1** *The objective function $f(\gamma)$ is strictly convex over its domain.*

*Proof.* The cost term $\langle \gamma, C \rangle$ is linear in $\gamma$, and the entropy term $H(\gamma)$ is strictly concave. Therefore, the regularized term $-\varepsilon H(\gamma)$ is strictly convex. The sum of a linear and strictly convex function remains strictly convex.

**Lemma 2** *The feasible set $\Pi(\mu, \nu) \subset \mathbb{R}^{n \times n}$ is non-empty, closed, convex, and compact.*

*Proof.* The constraints $\gamma \mathbf{1} = \mu$, $\gamma^\top \mathbf{1} = \nu$, and $\gamma \geq 0$ define a convex polytope. The entries of $\gamma$ are bounded within the interval $[0, 1]$ and their total sum is constant, ensuring compactness. The set is non-empty since $\gamma = \mu \nu^\top$ is a valid coupling that satisfies the marginal constraints.

**Proposition 1** *The entropy-regularized optimal transport problem*

$$\min_{\gamma \in \Pi(\mu, \nu)} \langle \gamma, C \rangle - \varepsilon H(\gamma) \tag{20}$$

*admits a unique optimal solution $\gamma^* \in \Pi(\mu, \nu)$.*

*Proof.* From Lemmas 1 and 2, the objective function is strictly convex and the feasible set is compact and non-empty. Therefore, by the fundamental theorem of convex optimization, a unique global minimizer exists.

**Proposition 2** *The unique minimizer $\gamma^*$ has the following closed-form structure:*

$$\gamma^* = \mathrm{diag}(u) \cdot K \cdot \mathrm{diag}(v), \quad K = \exp\left( -\frac{C}{\varepsilon} \right) \tag{21}$$

*for some positive scaling vectors $u, v \in \mathbb{R}_+^n$.*

*Proof.* Consider the Lagrangian:

$$\mathcal{L}(\gamma, \alpha, \beta) = \sum_{i,j} \gamma_{ij} C_{ij} - \varepsilon \sum_{i,j} \gamma_{ij} \log \gamma_{ij} + \sum_i \alpha_i \left( \mu_i - \sum_j \gamma_{ij} \right) + \sum_j \beta_j \left( \nu_j - \sum_i \gamma_{ij} \right) \tag{22}$$

Taking the derivative with respect to $\gamma_{ij}$ and setting it to zero yields:

$$\gamma_{ij} = \exp\left( \frac{\alpha_i + \beta_j - C_{ij}}{\varepsilon} - 1 \right) = u_i \cdot K_{ij} \cdot v_j \tag{23}$$

where $u_i = \exp\left( \frac{\alpha_i - 1}{\varepsilon} \right), \quad v_j = \exp\left( \frac{\beta_j}{\varepsilon} \right)$.

**Theorem 1** *Given a positive matrix $K \in \mathbb{R}_+^{n \times n}$ and marginal distributions $\mu, \nu \in \Delta_n$, there exist scaling vectors $u, v \in \mathbb{R}_+^n$ such that:*

$$\gamma = \mathrm{diag}(u) K \, \mathrm{diag}(v) \in \Pi(\mu, \nu) \tag{24}$$

*and the vectors $u, v$ can be computed via the following iterative updates:*

$$u^{(k+1)} = \frac{\mu}{K v^{(k)} + \delta} \tag{25}$$

$$v^{(k+1)} = \frac{\nu}{K^\top u^{(k+1)} + \delta} \tag{26}$$

*where $\delta > 0$ is a small constant to ensure numerical stability.*

### A.3 Training Strategy

For a clearer illustration of the training and inference process, we conclude them in Algorithm 2.

---

**Algorithm 2:** HPSERec

---

**Data:** $S_u$ for $u \in U$, learning rate $lr$, hyperparameters $\alpha, \beta, \tau, \eta, \gamma, L$.
**Result:** Last expert model parameters $\mathcal{W}_L$.

1   Get item sets $V_1, \ldots V_L$ by Algorithm 1;
2   **for** $i \leftarrow 1$ **to** $E$ (number of epochs) **do**
     /* Feedforward stage                                                      */
3      **for** $j \leftarrow 1$ **to** $L$ **do**
4          **if** $j > 1$ **then**
5              Calculate $\mathcal{L}_{KD}$ by Eq. (9);
6          **end**
7          Set $\mathcal{L}_{forw}$ by Eq. (13);
8          Apply Adam optimizer to $\mathcal{L}_{forw}$;
9          Perform back-propagation to $\mathcal{L}_{forw}$ getting gradients $\mathcal{G}$;
10         Update $\mathcal{W}_j$ based on $\mathcal{G}$;
11      **end**
     /* Feedback stage                                                          */
12     **for** $j \leftarrow L$ **to** $1$ **do**
13         Calculate $\mathcal{L}_{back}$ by Eq. (12);
14         Perform back-propagation to $\mathcal{L}_{back}$ getting gradients $\mathcal{G}$;
15         Update $\mathcal{W}_j$ based on $\mathcal{G}$;
16     **end**
17 **end**
18 **return** $\mathcal{W}_L$;

---

## B   Experimental Settings

In this section, we will refer to more details about the experimental settings.

### B.1   Dataset and Preprocessing

We compare HPSERec with baseline models using three real-world datasets from online services: Yelp, Amazon Beauty, and Amazon Music. For data pre-processing, we follow prior research by filtering out users with fewer than five interactions. Table 3 presents the statistics of the datasets after pre-processing.

### B.2   Backbone and Baseline

We compare our proposed method with the following baseline methods.

- **SASRec** employs a self-attention mechanism to model the user's entire interaction sequence and predict the next potential item for interaction.

Table 3: Statistics of datasets. #Int denotes interaction.

| Dataset | #Items | #Users | #Int | Avg $|S_u|$ |
|---------|--------|--------|--------|---------|
| Beauty | 57,289 | 52,204 | 394,908 | 5.6 |
| Yelp | 15,720 | 4,722 | 192,214 | 3.8 |
| Music | 20,356 | 20,165 | 132,595 | 5.1 |

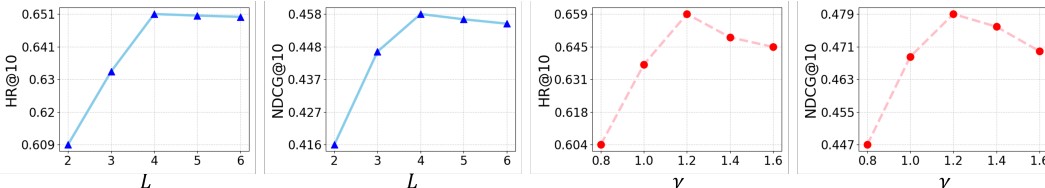

Figure 5: The hyper-parameter experiments on the weight of number of subsets $L$ and the weight of the regularization penalty $\gamma$. The result are based on the Music dataset with the SASRec model.

- **BERT4Rec** adopts the Cloze task to train a bidirectional model, extending SASRec for better sequential modeling.

- **CITIES** applies a self-attention mechanism, leveraging head items to improve the representation of tail items.

- **MELT** optimizes the representation of both long-tailed items and users, marking its first effort in this area.

- **RLMRec** leverages LLMs to generate user and item profiles, effectively capturing their interaction preferences.

- **LLMInit** uses LLM embeddings to initialize the embedding layer of the element in the SRS model.

- **LLM-ESR** enhances both long-tailed items and users through a dual view modeling framework combined with a retrieval-augmented self-distillation method.

## B.3 Implementation Details

We conduct all experiments on an Intel Core i7-11700KF platform with dual NVIDIA GeForce RTX 3090 (24GB) GPUs. Besides, the implementation is based on Python 3.8.19 and PyTorch 2.0.0. For the backbone SRS models, the number of GRU layers is set to 1 for GRU4Rec, while the number of self-attention layers is fixed at 2 for SASRec and Bert4Rec. Also, the dropout rate is 0.6 for Bert4Rec. We use the Adam optimizer for parameter optimization with a learning rate of $1 \times 10^{-4}$. The embedding size is 128 for all baselines, We choose the Adam as the optimizer. For Eq. (5), the default value of $\alpha$ is set to 0.6. For Eq. (6), the default value of $L$ is set to 4. For Eq. (7), the default value of $\gamma$ is set to 1.2. For Eq. (9), the default value of $\tau$ is set to 1.2. For Eq. (13), the default value of $\beta$ is set to 1.

## C More Experimental Results

### C.1 Hyper-parameter Analysis

Figure 5 illustrates the impact of several key hyperparameters on the performance of HPSERec. The hyperparameter $L$ controls the number of item subsets generated by the Distribution Balancing module. As $L$ increases from 2 to 6, the recommendation accuracy initially improves and then gradually declines. The suboptimal performance with a small $L$ can be attributed to the large representational disparity between subsets, while an excessively large $L$ may degrade the representation quality of upstream item subsets due to data sparsity. Regarding the regularization weight $\gamma$, which controls the penalty term in the partitioning objective, the optimal value is found to be 1.2. A smaller $\gamma$ results in imbalanced subset sizes, whereas a larger $\gamma$ overly suppresses the influence of item-level imbalance, thus impairing the quality of the learned partitions.

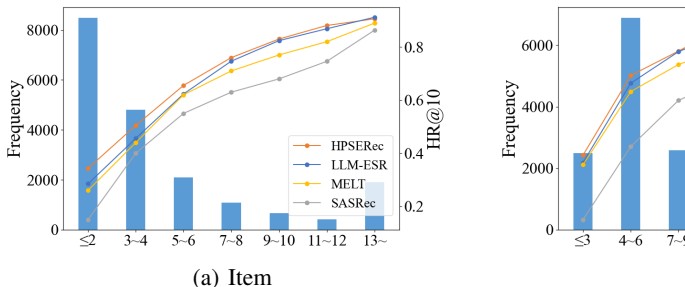

(a) Item        (b) User

Figure 6: The results of the proposed HPSERec and competing baselines in user and item groups. The results are based on the Music dataset with the SASRec model.

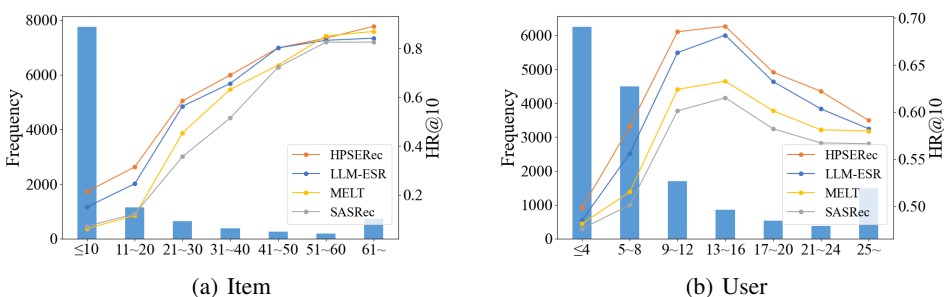

(a) Item        (b) User

Figure 7: The results of the proposed HPSERec and competing baselines in user and item groups. The results are based on the Yelp dataset with the SASRec model.

## C.2 Group Analysis

To further evaluate the effectiveness of HPSERec, we conduct group-wise performance analysis on two additional datasets, Music and Yelp. The results are presented in Figures 6 and Figures 7. From the results, we observe that the LLM-based framework consistently improves performance across all user and item groups. Moreover, HPSERec demonstrates a clear advantage under extreme long-tail scenarios, while maintaining competitive performance on head users and items, indicating its ability to enhance tail representations without compromising head accuracy.

## C.3 Ablation Studies

In this section, we present additional ablation experiments on two benchmark datasets, Yelp and Beauty, with results summarized in Table 4. Specifically, Row 1 corresponds to the baseline SASRec. Row 2 shows the performance when head and tail classes are partitioned using conventional frequency-based heuristics, and data augmentation is applied via the Feedforward module. Building on this, Row 3 incorporates the Feedback module to further enhance tail-side representations. Row 4 reflects the effect of applying our proposed Distribution Balancing module for data partitioning, coupled with Feedforward-based augmentation. Finally, Row 5 adds the Feedback module to Row 4, forming the complete HPSERec framework. Notably, although HPSERec does not include a dedicated module targeting tail users, we observe a consistent performance gain for this group. This is because user and item representations are embedded in the same vector space, and user embeddings are computed based on the items they interact with—thus, improving item representations inherently enhances user representations as well. Moreover, unlike many long-tail recommendation methods that favor tail performance at the cost of head accuracy, HPSERec achieves balanced improvements across head and tail users/items. The full model leads to consistent performance gains in both head and tail segments, demonstrating effective representation alignment and knowledge propagation across levels of data sparsity.

Table 4: The ablation study on the Yelp and Amazon Beauty dataset with SASRec as the backbone SRS model. The boldface refers to the highest score and the underline indicates the next best result of the models.

| Dataset | Row | DB | FF | FB | Overall | | Head User | | Tail User | | Head Item | | Tail Item | |
|---|---|---|---|---|---|---|---|---|---|---|---|---|---|---|
| | | | | | HR@10 | ND@10 | HR@10 | ND@10 | HR@10 | ND@10 | HR@10 | ND@10 | HR@10 | ND@10 |
| Yelp | 1 | | | | 0.5866 | 0.3536 | 0.5945 | 0.3585 | 0.5848 | 0.3591 | 0.8002 | 0.4888 | 0.0890 | 0.0386 |
| | 2 | | ✓ | | 0.6181 | 0.3799 | 0.6135 | 0.3784 | 0.6142 | 0.3807 | 0.8055 | 0.4941 | 0.1061 | 0.0582 |
| | 3 | | ✓ | ✓ | 0.6313 | 0.3974 | 0.6237 | 0.3897 | 0.6311 | 0.4008 | 0.8130 | 0.5031 | 0.1440 | 0.0874 |
| | 4 | ✓ | ✓ | | 0.6719 | 0.4225 | 0.6440 | **0.4065** | 0.6738 | 0.4238 | 0.8243 | 0.5206 | 0.3200 | 0.1797 |
| | 5 | ✓ | ✓ | ✓ | **0.6827** | **0.4231** | **0.6583** | 0.4027 | **0.6884** | **0.4280** | **0.8361** | **0.5261** | **0.3252** | **0.1832** |
| Beauty | 1 | | | | 0.4488 | 0.2861 | 0.5261 | 0.3611 | 0.4236 | 0.2690 | 0.7187 | 0.4815 | 0.1593 | 0.0856 |
| | 2 | | ✓ | | 0.4606 | 0.2977 | 0.5461 | 0.3873 | 0.4367 | 0.2875 | 0.7195 | 0.4844 | 0.1872 | 0.1083 |
| | 3 | | ✓ | ✓ | 0.4826 | 0.3209 | 0.5522 | 0.3907 | 0.4898 | 0.3216 | 0.7257 | 0.5026 | 0.2152 | 0.1439 |
| | 4 | ✓ | ✓ | | 0.5198 | 0.3417 | 0.5564 | 0.3965 | 0.5047 | 0.3402 | 0.7271 | 0.5125 | 0.2612 | 0.1712 |
| | 5 | ✓ | ✓ | ✓ | **0.5281** | **0.3665** | **0.5799** | **0.4148** | **0.5163** | **0.3557** | **0.7306** | **0.5229** | **0.3203** | **0.2060** |

# D Limitation

Two potential limitations of this work should be noted. First, the proposed framework involves a number of hyperparameters, and identifying optimal configurations for specific tasks may require considerable tuning effort. Second, the subset partitioning algorithm in the Distribution Balancing module incurs relatively high computational complexity, making it more suitable for scenarios with a moderate number of items. In practical applications with large-scale item sets, further optimization or integration with incremental learning strategies may be necessary.

