# OpenReview forum: "HPSERec: A Hierarchical Partitioning and Stepwise Enhancement Framework for Long-tailed Sequential Recommendation"
_NeurIPS.cc/2025/Conference — NeurIPS 2025 poster_

### Official Review · Reviewer_Cp2v · 2025-06-18

**Clarity:** 2
**Significance:** 3
**Originality:** 3
**Rating:** 4
**Confidence:** 3

**Summary:**

This paper investigates the sequential recommendation problem for long-tail items and proposes a stepwise enhancement approach. Specifically, it improves recommendation performance for long-tail items by progressively transferring knowledge among experts trained on different item groups.

**Questions:**

1.	What is the computational cost introduced by training experts? Could this be theoretically analyzed?
2.	Does the entire training process guarantee convergence? How is convergence determined?
3.	Does the stepwise data-increment method for training experts ensure diverse data distribution across different experts?
4.	Which expert does the model rely on for its final output?

**Ethical Concerns:**

["NO or VERY MINOR ethics concerns only"]

**Limitations:**

Limitations are not fully addressed. In order to improve the paper, some other ways to generate the sub-datasets to train different experts can be added.

**Paper Formatting Concerns:**

There is no formatting issue.

**Quality:**

3

**Strengths And Weaknesses:**

Strengths:
1.	The proposed idea of stepwise training experts at different levels is innovative, and its training methodology has broad generalizability.
2.	The appendix provides a theoretical proof for the feedback module.
3.	Experiments demonstrate significant performance improvements with the proposed method.

Weaknesses:
1.	The algorithm incurs higher computational costs. Beyond the notable cost of partitioning item groups, training experts also involves substantial overhead, which was not evaluated in the paper.
2.	The paper’s clarity could be improved. Some formulas are ambiguously expressed, including undefined symbols and a lack of explanations (e.g., Appendix Formula 18). Additionally, Equation (14) mentioned in the main text is not visible.
3.	The appendix (Section A.2) does not clearly explain the differences between the provided proof and the standard Sinkhorn algorithm.

---

> ### Author Rebuttal · Authors · 2025-07-31
>
> Thank you. We've made responses (Res) to all Questions (Q):
>
> ***Q1:** What is the computational cost introduced by training experts? Could this be theoretically analyzed?*
>
> **Res to Q1:** We sincerely thank you for raising this important question.
> In the Feedforward stage, using SASRec as the base recommender architecture, the overall computational complexity of expert training can be expressed as:
> $\mathcal{O}(E \cdot T^2 \cdot d \cdot L_T)$
> where $T$ denotes the sequence length, $d$ is the embedding dimension, $L_T$ is the number of Transformer layers, and $E$ is the number of expert modules. This reflects the fact that each expert is trained independently on a subset of data in a staged manner.
> In the Feedback stage, the cost is primarily associated with computing soft alignment between user representations via Sinkhorn optimal transport. Specifically,
> constructing the pairwise similarity matrix has complexity $\mathcal{O}(m^2 \cdot d)$.
> The Sinkhorn-Knopp algorithm, with t iterations, has complexity $\mathcal{O}(t \cdot m^2)$,
> resulting in an overall cost of
> $\mathcal{O}(m^2 \cdot (d + t))$, where $m$ is the batch size (typically 128 or 256), $d$ is the embedding dimension, and $t$ is the number of Sinkhorn iterations (usually between 10 and 20). Given the relatively small size of $m$, this cost remains manageable and GPU-friendly.
> Finally, in the inference stage, only the final expert is used for prediction. Hence, the inference complexity is $\mathcal{O}(T^2 \cdot d \cdot L_T)$, which is equivalent to that of a standard SASRec model, ensuring no additional latency at test time.
>
> In summary, the overall training complexity is $\mathcal{O}(E \cdot T^2 \cdot d \cdot L_T) + \mathcal{O}(m^2 \cdot (d + t)),$
> and the inference complexity is $\mathcal{O}(T^2 \cdot d \cdot L_T)$.
>
>
> ***Q2:** Does the entire training process guarantee convergence? How is convergence determined?*
>
> **Res to Q2:** Thank you for raising this important question. Due to the complexity of our model, which involves multiple expert networks, knowledge distillation, and the Sinkhorn optimal transport mechanism, we have not provided a formal theoretical proof of convergence in the paper. However, empirical evidence shows that the training process of HPSERec is stable and exhibits clear convergent behavior. For example, on the Music dataset, HPSERec typically converges within approximately 20 epochs, while on the Yelp dataset, convergence is generally achieved within around 15 epochs.
> We employ the following combined strategies to determine model convergence:
> 1. Validation metrics (e.g., HR@10, NDCG@10) are computed after each epoch. Early stopping is triggered if no improvement is observed for a consecutive number of epochs (default: 5).
> 2. The overall training loss (encompassing both cross-entropy and feedback losses) consistently exhibits stable descent and eventual convergence across multiple experimental runs.
>
> ***Q3:** Does the stepwise data-increment method for training experts ensure diverse data distribution across different experts?*
>
> **Res to Q3:** Thank you for your question. In our method, each expert receives a different subset of user interaction sequences and is assigned a distinct responsibility. Specifically, the first expert is trained only on long-tail items, the second expert on long-tail plus mid-tail items, and so on, with the final expert receiving all items. Accordingly, the first expert focuses on learning users’ long-tail interests, the second expert on mid-tail interests, and the final expert on users’ global preferences. This diversity in training data across experts inherently ensures that each expert captures a different aspect of the user-item distribution.
>
>
> ***Q4:** Which expert does the model rely on for its final output?*
>
> **Res to Q4:** Thank you for raising this question. As explained in our response to the previous question, the final expert is responsible for learning the complete user interest representation. Therefore, the model relies on the output of the last expert for its final prediction.

---

> > ### Comment · Reviewer_Cp2v · 2025-08-05
> >
> > Thank you for your response, especially the analysis on  computational cost.

---

> > > ### Author Response · Authors · 2025-08-06
> > >
> > > Thank you very much for your valuable feedback and constructive suggestions on our work. We are pleased to have addressed your concerns.

---

### Official Review · Reviewer_2hmm · 2025-06-27

**Clarity:** 3
**Significance:** 4
**Originality:** 4
**Rating:** 5
**Confidence:** 5

**Summary:**

This paper presents HPSERec, a framework for long-tailed sequential recommendation that mitigates item popularity bias. It features: (1) hierarchical item partitioning via a dynamic imbalance metric; (2) stepwise expert training with knowledge distillation; and (3) a feedback module using Sinkhorn optimal transport for global alignment. Experiments on three datasets demonstrate superior tail-item performance over traditional and LLM-based baselines without sacrificing head-item accuracy.

**Questions:**

1. The proposed method trains user interests starting from tail interests in the feedforward module until complete user interests are learned. However, is the reverse approach more reasonable?
2. Why does this paper not include comparisons with classical long-tail recommendation methods, such as LightGCN?
3. Is there a potential error in the definition of data imbalance (Equation 5)? Could the authors clarify the meanings of the two exponential terms (α and 1/(1−α)) separately?
4. While the paper primarily addresses the long-tail distribution of known items, how does the proposed method perform on entirely cold-start items?

**Ethical Concerns:**

["NO or VERY MINOR ethics concerns only"]

**Final Justification:**

My concerns have been well addressed. I would like to keep my positive score.

**Limitations:**

yes

**Quality:**

3

**Strengths And Weaknesses:**

Strengths:
1. This work proposes a hierarchical dynamic partitioning strategy that utilizes a data imbalance metric and dynamic programming algorithm to replace traditional fixed-ratio partitioning methods, demonstrating superior adaptability to varying skewed distributions across different datasets.
2. The stepwise enhancement approach progressively improves tail item representations through a Mixture-of-Experts network architecture, combined with knowledge distillation, effectively mitigating the issue of negative transfer in representation learning.
3. The study makes a theoretical contribution by being the first to decompose the long-tail problem into multi-level subtasks, presenting a comprehensive framework that integrates metric design, partitioning, distillation, and alignment.

Weaknesses:
1. The dynamic programming-based partitioning algorithm exhibit computational inefficiency when applied to extremely large-scale item sets, requiring further optimization or approximation methods for practical deployment.
2. While the method indirectly improves performance for tail users through item representation enhancement, the lack of dedicated modules specifically addressing user-side sparsity could potentially limit its effectiveness for extreme cold-start user scenarios.
3. The alternating training strategy significantly increases training complexity, potentially affecting the method's scalability and practical implementation efficiency.

---

> ### Author Rebuttal · Authors · 2025-07-31
>
> Thank you. We've made responses (Res) to all Questions (Q):
>
> ***Q1:** The proposed method trains user interests starting from tail interests in the feedforward module until complete user interests are learned. However, is the reverse approach more reasonable?*
>
> **Res to Q1:** Thank you for raising this insightful question. This issue was indeed considered during our experimental exploration. Our empirical results show that starting from tail interests achieves superior performance.
> This observation can be interpreted as follows: In the feedforward module, beginning from head items allows the model to learn well-established knowledge, which is then propagated to the global user representation. While this seems plausible, it presents a limitation in the feedback stage. Since head interests are already well trained, using the global representation to supervise them yields marginal gains.
> In contrast, our proposed approach begins with tail interests in the feedforward module and progressively transfers this knowledge upward. We hypothesize that this process implicitly assigns more importance to tail interests, encouraging the global user representation to better encode information from the long tail. Subsequently, in the feedback module, the global representation guides the refinement of tail interests, delivering additional supervision and significantly enhancing their accuracy. This targeted reinforcement helps to overcome the inherent sparsity and noise associated with tail data, leading to more robust overall performance.
>
> ***Q2:** Why does this paper not include comparisons with classical long-tail recommendation methods, such as LightGCN?*
>
> **Res to Q2:** Thank you for raising this important question. As demonstrated in many related studies [2,6], the accuracy of several classical long-tail recommendation methods, such as LightGCN, is generally lower than that of the more advanced baselines we compared against in this paper. To further strengthen our experimental evaluation, we will include two recent long-tail recommendation methods, named ASReP and INSERT, in the comparative experiments in the final version of the paper.
> |  Model  | Dataset | Overall |         | Head user |         | Tail user |         | Head item |        | Tail item |         |
> |:-------:|:-------:|:-------:|:-------:|:---------:|:-------:|:---------:|:-------:|-----------|--------|-----------|---------|
> |         |         | HR@10      | NDCG@10    | HR@10        | NDCG@10    | HR@10        | NDCG@10    | HR@10       | NDCG@10   | HR@10        | NDCG@10    |
> | HPSERec |  Music  | 0.6592  | 0.4786  | 0.7144    | 0.5069  | 0.6428    | 0.4701  | 0.8989    | 0.6806 | 0.4425    | 0.2959  |
> | ASEeP   |         | 0.5448  | 0.3573  | 0.6573    | 0.4591  | 0.4943    | 0.3496  | 0.8254    | 0.6225 | 0.2637    | 0.1559  |
> | INSERT  |         | 0.4414  | 0.2774  | 0.4845    | 0.3103  | 0.4174    | 0.2607  | 0.8455    | 0.5385 | 0.0776    | 0.031   |
> | HPSERec |   Yelp  | 0.6827  | 0.4231  | 0.6583    | 0.4027  | 0.6884    | 0.428   | 0.8361    | 0.5261 | 0.3252    | 0.1832  |
> | ASEeP   |         | 0.5307  | 0.3032  | 0.5141    | 0.2899  | 0.5277    | 0.2873  | 0.6821    | 0.3894 | 0.0161    | 0.0057  |
> | INSERT  |         | 0.4873  | 0.2746  | 0.4728    | 0.2543  | 0.4895    | 0.2774  | 0.6264    | 0.3418 | 0.0177    | 0.0059  |
> | HPSERec |  Beauty | 0.5281  | 0.3665  | 0.5799    | 0.4148  | 0.5163    | 0.3557  | 0.7306    | 0.5229 | 0.3203    | 0.2060  |
> | ASEeP   |         | 0.4674  | 0.3108  | 0.5375    | 0.3766  | 0.4515    | 0.284   | 0.7531    | 0.5113 | 0.1764    | 0.0899  |
> | INSERT  |         | 0.4091  | 0.2682  | 0.4371    | 0.2672  | 0.4018    | 0.2470  | 0.7503    | 0.4511 | 0.0275    | 0.0101  |
>
> [2] K. Kim, D. Hyun, S. Yun, and C. Park, “Melt: Mutual enhancement of long-tailed user and
> item for sequential recommendation,” in Proceedings of the 46th international ACM SIGIR
> conference on Research and development in information retrieval, 2023, pp. 68–77.
>
> [6]Z. Liu, Z. Fan, Y. Wang, and P. S. Yu, “Augmenting sequential recommendation with pseudoprior items via reversely pre-training transformer,” in Proceedings of the 44th international ACM SIGIR conference on Research and development in information retrieval, 2021, pp. 1608–1612.
>
> ***Q3:** Is there a potential error in the definition of data imbalance (Equation 5)? Could the authors clarify the meanings of the two exponential terms (α and 1/(1−α)) separately?*
>
> **Res to Q3:** Thank you for raising this important question. The definition in Equation (5) is essentially derived from a variant of Rényi entropy, which is designed to quantify the unevenness of interaction frequency distributions within each subset. In our context, it serves as a measure of the “entropy-based imbalance” for each partitioned item group.
>
> 1) The parameter α controls the sensitivity of the metric to the distribution of probabilities. When α -> 0, all items are treated nearly equally, resembling a measure of set cardinality. As α -> \infty, the metric becomes increasingly dominated by the highest-frequency items, thus emphasizing head items in a long-tail distribution. A smaller α encourages more uniform subset splits, while a larger α highlights skewed distributions. In essence, α determines the smoothness and focus of the entropy measure.
> 2) 1/1-α is not an independent parameter but a structural component of the Rényi entropy definition. It acts as a power transformation to normalize the scale of \sum p_i^α across different values of α, ensuring that the resulting measure remains comparable across settings. Its purpose is to map the aggregated value back into the entropy space, where higher values indicate lower imbalance and greater distributional uniformity.
>
> ***Q4:** While the paper primarily addresses the long-tail distribution of known items, how does the proposed method perform on entirely cold-start items?*
>
> **Res to Q4:** Thank you for raising this important question. The key issue addressed in this paper is the long-tail problem, which, although related to the cold-start problem, has a distinct definition and focus. The long-tail problem arises from the imbalance in user-item interactions, which results in suboptimal recommendations for users and limited exposure for less popular items. In contrast, the cold-start problem refers to the challenge of making recommendations when users or items have little to no historical interaction data. In our future work, we plan to explore how the proposed framework can be extended to effectively handle cold-start scenarios.

---

> > ### Comment · Reviewer_2hmm · 2025-08-05
> >
> > Thank you for your detail response. My concerns have been well addressed. I would like to keep my positive score.

---

> > > ### Author Response · Authors · 2025-08-06
> > >
> > > Thank you very much for your valuable feedback. We sincerely appreciate your time and constructive suggestions.

---

### Official Review · Reviewer_zHNT · 2025-07-01

**Clarity:** 3
**Significance:** 2
**Originality:** 2
**Rating:** 4
**Confidence:** 3

**Summary:**

This paper proposes HPSERec, a method that splits items into balanced groups and uses stepwise knowledge transfer to better recommend rare (tail) items. It improves long-tail recommendation accuracy while keeping overall performance strong.

**Questions:**

Regarding the division of head and tail, it seems that for different data sets, a simpler strategy can be used to divide them through statistical information. The division method proposed by the author is reasonable, but is it necessary? Could the authors provide comparative experiments using simpler or naive grouping strategies to better demonstrate the necessity and effectiveness of their proposed partitioning approach?

**Ethical Concerns:**

["NO or VERY MINOR ethics concerns only"]

**Final Justification:**

The authors have addressed my concerns. I will raise my score of this paper to 4.

**Limitations:**

yes

**Quality:**

3

**Strengths And Weaknesses:**

Pros:
1. Clear motivation, focusing on a problem of practical significance.
2. Good Writing.
3. The method is reasonable and provides theoretical support.

Cons:
1. Regarding the division of head and tail, it seems that for different data sets, a simpler strategy can be used to divide them through statistical information. The division method proposed by the author is reasonable, but the necessity needs further discussion.
2. Regarding the grouping method, there is also a lack of comparative experiments with some naive strategy groupings.
3. The author focuses on the long-tail problem, but in the selection of baselines, there are not enough methods that focus on the long-tail problem for comparison.
4. The experiment in Figure 4 is difficult to understand, and the correspondence between the two axes and the figure should be more clearly stated. At the same time, it seems that the performance trends of different methods in different groups are consistent, which does not seem to clearly show the advantages of the author's method for long-tail problems.
5. The method is relatively complex, and it is recommended to add experiments to demonstrate its scalability.

---

> ### Author Rebuttal · Authors · 2025-07-31
>
> Thank you. We've made responses (Res) to all weaknesses (W) and Questions (Q):
>
> ***W1, W2 & Q1:** Regarding the division of head and tail, it seems that for different data sets, a simpler strategy can be used to divide them through statistical information. The division method proposed by the author is reasonable, but is it necessary? Could the authors provide comparative experiments using simpler or naive grouping strategies to better demonstrate the necessity and effectiveness of their proposed partitioning approach?*
>
> **Response to W1, W2 & Q1:** We sincerely thank the reviewer for this insightful suggestion. While simpler partitioning strategies do exist, they tend to compromise overall model accuracy, which is the central focus of this work. In fact, we carefully considered multiple partitioning strategies during the design of our method, and experimental results consistently showed that these alternatives underperformed compared to our proposed approach. To further evaluate the necessity and effectiveness of our method, we conducted experiments with two additional variants: 1) HPSERec-QSplit adopts a simpler quantile-based partitioning strategy that statically divides items based on their interaction counts. This comparison helps evaluate whether the use of more complex partitioning algorithms is justified in practice. 2) HPSERec-DPPrune incorporates a pruning-based optimization into our original dynamic programming framework. While this variant does not guarantee finding the global optimum, it substantially improves computational efficiency.
>
> The corresponding experimental results are presented below.
>
> |      Model      | Dataset | Overall |         | Head user |         | Tail user |        | Head item |        | Tail item |         |
> |:---------------:|:-------:|:-------:|:-------:|:---------:|:-------:|:---------:|:------:|-----------|--------|-----------|---------|
> |                 |         | HR@10      | NDCG@10    | HR@10        | NDCG@10    | HR@10        | NDCG@10   | HR@10        | NDCG@10  | HR@10        | NDCG@10    |
> | HPSERec         |  Music  | **0.6592**  | **0.4786**  | **0.7144**    | **0.5069**  | **0.6428**    | **0.4701** | **0.8989**    | **0.6806** | **0.4425**    | **0.2959**  |
> | HPSERec-QSplit  |         | 0.5847  | 0.4151  | 0.6717    | 0.4682  | 0.5588    | 0.3993 | 0.8442    | 0.6287 | 0.3501    | 0.2220   |
> | HPSERec-DPPrune |         | 0.6280   | 0.4423  | 0.7112    | 0.5028  | 0.6088    | 0.4311 | 0.8943    | 0.6798 | 0.3871    | 0.2407  |
> | HPSERec         |   Yelp  | **0.6827**  | **0.4231**  | **0.6583**    | **0.4027**  | **0.6884**    | **0.4280**  | **0.8361**    | **0.5261** | **0.3252**    | **0.1832**  |
> | HPSERec-QSplit  |         | 0.6457  | 0.3940  | 0.6458    | 0.3910  | 0.6466    | 0.3947 | 0.8236    | 0.5139 | 0.2309    | 0.1146  |
> | HPSERec-DPPrune |         | 0.6585  | 0.4063  | 0.6542    | 0.3988  | 0.6594    | 0.4082 | 0.8282    | 0.5233 | 0.2629    | 0.1339  |
> | HPSERec         |  Beauty | **0.5281**  | **0.3665**  | **0.5799**    | **0.4148**  | **0.5163**    | **0.3557** | **0.7306**    | **0.5229** | **0.3203**    | **0.2060**  |
> | HPSERec-QSplit  |         | 0.4263  | 0.2673  | 0.4866    | 0.3229  | 0.4125    | 0.2546 | 0.7056    | 0.4556 | 0.1395    | 0.0741  |
> | HPSERec-DPPrune |         | 0.4509  | 0.2982  | 0.5181    | 0.3592  | 0.4356    | 0.2844 | 0.6967    | 0.4701 | 0.1986    | 0.1219  |
>
> ***W3:** The author focuses on the long-tail problem, but in the selection of baselines, there are not enough methods that focus on the long-tail problem for comparison.*
>
> **Res to W3:** Thank you for highlighting this limitation. Among the seven baselines we included in our current experiments, three methods, CITIES, MELT, and LLM-ESR, are specifically designed to address the long-tail recommendation problem. In the final version of the paper, we will further enrich the comparative evaluation by including additional long-tail-focused baselines, namely ASReP and INSERT, to provide a more comprehensive assessment.
>
> The corresponding experimental results are presented below.
>
> |  Model  | Dataset | Overall |         | Head user |         | Tail user |         | Head item |        | Tail item |         |
> |:-------:|:-------:|:-------:|:-------:|:---------:|:-------:|:---------:|:-------:|-----------|--------|-----------|---------|
> |         |         | HR@10      | NDCG@10    | HR@10        | NDCG@10    | HR@10        | NDCG@10    | HR@10       | NDCG@10   | HR@10        | NDCG@10    |
> | HPSERec |  Music  | **0.6592**  | **0.4786**  | **0.7144**    | **0.5069**  | **0.6428**    | **0.4701**  | **0.8989**    | **0.6806** | **0.4425**    | **0.2959**  |
> | ASEeP   |         | 0.5448  | 0.3573  | 0.6573    | 0.4591  | 0.4943    | 0.3496  | 0.8254    | 0.6225 | 0.2637    | 0.1559  |
> | INSERT  |         | 0.4414  | 0.2774  | 0.4845    | 0.3103  | 0.4174    | 0.2607  | 0.8455    | 0.5385 | 0.0776    | 0.0310   |
> | HPSERec |   Yelp  | **0.6827**  | **0.4231**  | **0.6583**    | **0.4027**  | **0.6884**    | **0.4280**   | **0.8361**    | **0.5261** | **0.3252**    | **0.1832**  |
> | ASEeP   |         | 0.5307  | 0.3032  | 0.5141    | 0.2899  | 0.5277    | 0.2873  | 0.6821    | 0.3894 | 0.0161    | 0.0057  |
> | INSERT  |         | 0.4873  | 0.2746  | 0.4728    | 0.2543  | 0.4895    | 0.2774  | 0.6264    | 0.3418 | 0.0177    | 0.0059  |
> | HPSERec |  Beauty | **0.5281**  | **0.3665**  | **0.5799**    | **0.4148**  | **0.5163**    | **0.3557**  | **0.7306**    | **0.5229** | **0.3203**    | **0.2060**  |
> | ASEeP   |         | 0.4674  | 0.3108  | 0.5375    | 0.3766  | 0.4515    | 0.284   | 0.7531    | 0.5113 | 0.1764    | 0.0899  |
> | INSERT  |         | 0.4091  | 0.2682  | 0.4371    | 0.2672  | 0.4018    | 0.2470  | 0.7503    | 0.4511 | 0.0275    | 0.0101  |
>
> ***W4:** The experiment in Figure 4 is difficult to understand, and the correspondence between the two axes and the figure should be more clearly stated. At the same time, it seems that the performance trends of different methods in different groups are consistent, which does not seem to clearly show the advantages of the author's method for long-tail problems.*
>
> **Res to W4:** Thank you for your valuable suggestion. We acknowledge that Figure 4 lacks a clear explanation in its current form. In this figure, we split users and items into seven groups based on sequence length and popularity, respectively. The bar chart represents the number of users or items in each group, while the line chart indicates the recommendation performance of each method within the corresponding group.
>
> It is true that all methods exhibit a consistent performance trend, with weaker results on tail data and stronger results on head data, which is a common characteristic in long-tail recommendation scenarios. However, closer inspection reveals distinctive behaviors among the models: for instance, MELT performs well on head items but suffers significantly on tail items. In contrast, our proposed HPSERec achieves the best performance across all user groups, demonstrating its ability to provide consistent and high-quality recommendations for both head and tail users.
>
> In the final version of the paper, we will offer a more precise explanation of both axes and the comparative insights it conveys, to ensure greater clarity and stronger support for our claims.
>
>
> ***W5:** The method is relatively complex, and it is recommended to add experiments to demonstrate its scalability.*
>
> **Res to W5:** Thank you for your suggestion. We acknowledge that the current version of the paper lacks a thorough discussion of the scalability of our proposed method. In the present submission, we have evaluated the integration of our framework with SASRec. To address this concern, we will include additional experiments in the final version by incorporating our framework into FMLP, thereby providing a more comprehensive analysis of the method’s scalability and adaptability to different backbone architectures.

---

> > ### Comment · Reviewer_zHNT · 2025-08-05
> >
> > The authors have addressed my concerns. I will raise my score of this paper to 4.

---

> > > ### Author Response · Authors · 2025-08-06
> > >
> > > We sincerely appreciate your valuable feedback and are glad that our revisions have addressed your concerns. Thank you again for recognizing our efforts through the improved scores.

---

### Official Review · Reviewer_hCvg · 2025-07-02

**Clarity:** 3
**Significance:** 3
**Originality:** 3
**Rating:** 5
**Confidence:** 4

**Summary:**

The paper introduces HPSERec (Hierarchical Progressive Stepwise Enhancement for Long-Tail Recommendation), a novel framework designed to address the long-tail problem in sequential recommendation systems. HPSERec integrates hierarchical dynamic partitioning, progressive knowledge distillation, and optimal transport-based feedback alignment to improve the modeling of sparse and imbalanced user-item interactions. By leveraging a data-driven distribution balancing module and multi-level expert networks, the framework adapts to varying degrees of item popularity while avoiding negative transfer. Unlike LLM-enhanced approaches, HPSERec achieves robust performance without relying on large language models, making it more efficient and scalable for real-world deployment.

**Questions:**

1. The authors note in the appendix that the proposed method is only suitable for small-scale datasets, but fail to clarify its potential industrial application scenarios. Could the authors elaborate on how this approach might be adapted or scaled for real-world industrial deployment?

2. Although the proposed alternating training strategy—optimizing downstream experts during the feedforward moduleand fine-tuning upstream experts in the feedback module—significantly improves model performance, does the backpropagation-based update of upstream expert parameters during the feedback module risk disrupting the already stabilized parameters learned in the feedforward module?

**Ethical Concerns:**

["NO or VERY MINOR ethics concerns only"]

**Final Justification:**

I have read the author's responses to my concerns in their rebuttal. Thank you for the efforts of the authors and the AC, which have helped me further reflect on the score evaluation.

**Quality:**

3

**Strengths And Weaknesses:**

Strengths:

1. The work proposes a principled hierarchical decomposition of long-tail distributions, moving beyond simplistic head/tail dichotomy to provide a more granular and theoretically grounded approach to sparsity mitigation in recommendation systems.

2. A novel optimal transport-assisted feedback alignment mechanism is introduced, which leverages Sinkhorn-based distribution matching to theoretically guarantee feature consistency across different hierarchical subsets, ensuring more robust knowledge transfer.

3. Comprehensive experimental validation across multiple benchmark datasets demonstrates the framework's effectiveness, while rigorous ablation studies provide valuable insights into the contribution of individual components to the overall performance.

4. Compared to LLM-dependent approaches like RLMRec, HPSERec achieves comparable performance while operating within traditional recommendation architectures, making it significantly more scalable and practical for real-world deployment.

Weaknesses:

1. The cascaded computation in the multi-layer MoE (Mixture of Experts) architecture increases both memory consumption and inference latency, which potentially impact the real-time responsiveness requirements of online recommendation services.

2. The theoretical foundation of the optimal transport component assumes ideal uniform distributions, an assumption that does not fully align with real-world user behavior data that often exhibits strong clustering effects and complex distribution patterns.

---

> ### Author Rebuttal · Authors · 2025-07-31
>
> Thank you. We've made responses (Res) to all Questions (Q):
>
> ***Q1:** The authors note in the appendix that the proposed method is only suitable for small-scale datasets, but fail to clarify its potential industrial application scenarios. Could the authors elaborate on how this approach might be adapted or scaled for real-world industrial deployment?*
>
>
> **Res to Q1:** Thank you for raising this important question. This method can mainly be divided into two parts: dataset division and recommendation framework. Although the algorithm complexity of dataset division is relatively high, in the industrial field, it is acceptable for an item pool in the tens of millions. As for the recommendation framework part, although the training process involves multiple expert networks, in the inference stage, we only use the last expert network. In future research, we will combine it with incremental learning algorithms to further enhance the practicality of this method.
>
>
> ***Q2:** Although thposed alternating training strategy—optimizing downstream experts during the feedforward moduleand fine-tuning upstream experts in the feedback module—significantly improves model performance, does the backpropagation-based update of upstream expert parameters during the feedback module risk disrupting the already stabilized parameters learned in the feedforward module?*
>
> **Res to Q2:** Thank you for the insightful question. While it is reasonable to be concerned that updating upstream experts via backpropagation may cause representation drift, our framework is specifically designed to mitigate this risk. The updates in the feedback module are guided by an optimal transport plan that ensures distribution-consistent fine-tuning rather than disruptive re-learning. Empirically, the feedback module consistently improves performance across head and tail users/items, indicating that it enhances rather than destabilizes upstream representations.

---

> > ### Comment · Reviewer_hCvg · 2025-08-07
> >
> > Thank you for the rebuttal. I have reviewed all the responses and reviews, and I will stand by my original score as I believe it is a fair assessment of this paper.

---

> > > ### Author Response · Authors · 2025-08-08
> > >
> > > Thank you very much for your insightful and constructive feedback. We sincerely appreciate your time and the thoughtful suggestions you have provided.

---

### Official Review · Reviewer_SjUj · 2025-07-07

**Clarity:** 3
**Significance:** 2
**Originality:** 3
**Rating:** 4
**Confidence:** 3

**Summary:**

This paper addresses the long-tail problem in Sequential Recommendation Systems (SRS) by proposing HPSERec, a framework combining (1) an imbalance-aware hierarchical partitioning of items, (2) a feedforward knowledge distillation mechanism to enhance tail item representations, and (3) a feedback module using Sinkhorn-based optimal transport to align user representations across different expert levels. The framework aims to boost the performance on tail items and users while maintaining recommendation quality for head items. Extensive experiments on three real-world datasets show that HPSERec consistently outperforms both traditional and large language model-enhanced baselines in overall performance and long-tail recommendation accuracy.

**Questions:**

- How sensitive is the performance of HPSERec to the number of partitions (L)? Have you explored adaptive or learned partitioning beyond static imbalance-based splitting?

**Ethical Concerns:**

["NO or VERY MINOR ethics concerns only"]

**Limitations:**

The key limitations are summarized in the weaknesses section above.

**Paper Formatting Concerns:**

No concerns.

**Quality:**

3

**Strengths And Weaknesses:**

Strengths

- The hierarchical design with imbalance-based partitioning, stepwise knowledge transfer via distillation, and global alignment via optimal transport is novel, well-integrated, and mathematically sound, with clear theoretical grounding.

- The proposed method shows consistent improvements over competitive baselines, including recent LLM-based methods, across multiple datasets and scenarios (tail users/items, overall ranking), with solid ablations validating each module’s contribution.

Weaknesses

- While the proposed distribution balancing algorithm improves over fixed-ratio partitioning, it still assumes static, discrete partitions of items and users. The method does not address temporal dynamics, gradual shifts in item popularity, or cross-group interactions, which are common in real-world recommendation scenarios. This limits the generalizability of the approach beyond the datasets and static environments tested.

- The use of multiple expert models, stepwise distillation, and computationally expensive Sinkhorn optimal transport introduces non-negligible computational overhead. The paper lacks a detailed analysis of training and inference efficiency, particularly on large-scale datasets or in real-time scenarios.

---

> ### Author Rebuttal · Authors · 2025-07-31
>
> Thank you. We've made responses (Res) to all weaknesses (W) and Questions (Q):
>
> ***W1:** While the proposed distribution balancing algorithm improves over fixed-ratio partitioning, it still assumes static, discrete partitions of items and users. The method does not address temporal dynamics, gradual shifts in item popularity, or cross-group interactions, which are common in real-world recommendation scenarios. This limits the generalizability of the approach beyond the datasets and static environments tested.*
>
> **Res to W1:** Thank you for your insightful question. Our method is not intended for modeling temporal dynamics or behavioral evolution, but rather focuses on constructing a more reasonable data partitioning structure to enable subsequent models to better utilize tail data. In our future work, we can incorporate the time factor by combining "sliding window partitioning" or "incremental re-grouping" methods, and integrate incremental learning recommendation algorithms to further enhance the practicality of the method.
>
> ***W2:** The use of multiple expert models, stepwise distillation, and computationally expensive Sinkhorn optimal transport introduces non-negligible computational overhead. The paper lacks a detailed analysis of training and inference efficiency, particularly on large-scale datasets or in real-time scenarios.*
>
> **Res to W2:** Thank you for your valuable comment. We agree that computational overhead is a critical factor in assessing the practicality of recommendation systems. Compared to recent long-tail recommendation methods that incorporate large language models, our method introduces significantly lower computational complexity.
>
> The training complexity of our method is
> $\mathcal{O}(E \cdot T^2 \cdot d \cdot L_T) + \mathcal{O}(m^2 \cdot (d + t)),$
> where $E$ is the number of expert modules, $T$ is the sequence length, $d$ is the embedding dimension, $L_T$ is the number of Transformer layers, $m$ is the user batch size, and $t$ is the number of Sinkhorn iterations.
>
> The inference complexity is $\mathcal{O}(T^2 \cdot d \cdot L_T),$
> which matches that of standard Transformer-based recommenders, as only the final expert is used during inference.
> We will include the complete theoretical analysis of computational complexity in the appendix of the final version of the paper.
>
> ***Q1:** How sensitive is the performance of HPSERec to the number of partitions (L)?*
>
> **Res to Q1:** We thank you for your insightful question. In the hyperparameter analysis section of our paper (Appendix B.2), we systematically investigate the impact of the number of partitions L on model performance. Specifically, we evaluate the model on the Yelp and Beauty datasets by varying L from 2 to 6, and observe the following trends:
>
> 1. When L ranges from 2 to 4, the model exhibits notable improvements in HR@10 and NDCG@10, indicating that appropriately partitioning the item space facilitates tail representation enhancement and alleviates data imbalance.
> 2. When L > 4, the performance gains become marginal or slightly degrade. This is primarily due to over-segmentation, which results in fewer interactions per upstream subset. Consequently, the representation quality deteriorates, diminishing the effectiveness of knowledge distillation.
>
> Overall, HPSERec exhibits moderate sensitivity to the choice of $L$, but maintains stable performance within a reasonable range, making it practical and easy to tune in real-world applications.
>
> ***Q2:** Have you explored adaptive or learned partitioning beyond static imbalance-based splitting?*
>
> **Res to Q2:** Thank you for your thoughtful question. We have indeed explored adaptive or learned partitioning methods, such as incorporating clustering losses to guide subset construction via deep learning. However, these approaches did not yield satisfactory results and introduced substantial computational overhead to the overall framework.
> To further evaluate the necessity and effectiveness of our method, we conducted experiments with two additional variants: 1) HPSERec-QSplit adopts a simpler quantile-based partitioning strategy that statically divides items based on their interaction counts. This comparison helps evaluate whether the use of more complex partitioning algorithms is justified in practice. 2) HPSERec-DPPrune incorporates a pruning-based optimization into our original dynamic programming framework. While this variant does not guarantee finding the global optimum, it substantially improves computational efficiency.
> The corresponding experimental results are presented below.
>
> |      Model      | Dataset | Overall |         | Head user |         | Tail user |        | Head item |        | Tail item |         |
> |:---------------:|:-------:|:-------:|:-------:|:---------:|:-------:|:---------:|:------:|-----------|--------|-----------|---------|
> |                 |         | HR@10      | NDCG@10    | HR@10        | NDCG@10    | HR@10        | NDCG@10   | HR@10        | NDCG@10  | HR@10        | NDCG@10    |
> | HPSERec         |  Music  | **0.6592**  | **0.4786**  | **0.7144**    | **0.5069**  | **0.6428**    | **0.4701** | **0.8989**    | **0.6806** | **0.4425**    | **0.2959**  |
> | HPSERec-QSplit  |         | 0.5847  | 0.4151  | 0.6717    | 0.4682  | 0.5588    | 0.3993 | 0.8442    | 0.6287 | 0.3501    | 0.2220   |
> | HPSERec-DPPrune |         | 0.6280   | 0.4423  | 0.7112    | 0.5028  | 0.6088    | 0.4311 | 0.8943    | 0.6798 | 0.3871    | 0.2407  |
> | HPSERec         |   Yelp  | **0.6827**  | **0.4231**  | **0.6583**    | **0.4027**  | **0.6884**    | **0.4280**  | **0.8361**    | **0.5261** | **0.3252**    | **0.1832**  |
> | HPSERec-QSplit  |         | 0.6457  | 0.3940  | 0.6458    | 0.3910  | 0.6466    | 0.3947 | 0.8236    | 0.5139 | 0.2309    | 0.1146  |
> | HPSERec-DPPrune |         | 0.6585  | 0.4063  | 0.6542    | 0.3988  | 0.6594    | 0.4082 | 0.8282    | 0.5233 | 0.2629    | 0.1339  |
> | HPSERec         |  Beauty | **0.5281**  | **0.3665**  | **0.5799**    | **0.4148**  | **0.5163**    | **0.3557** | **0.7306**    | **0.5229** | **0.3203**    | **0.2060**  |
> | HPSERec-QSplit  |         | 0.4263  | 0.2673  | 0.4866    | 0.3229  | 0.4125    | 0.2546 | 0.7056    | 0.4556 | 0.1395    | 0.0741  |
> | HPSERec-DPPrune |         | 0.4509  | 0.2982  | 0.5181    | 0.3592  | 0.4356    | 0.2844 | 0.6967    | 0.4701 | 0.1986    | 0.1219  |
>
> Although the time complexity of these schemes is not as high as that of the method described in the text, the focus of this study is on accuracy, so we have abandoned the other methods.

---

### Decision · Program_Chairs · 2025-09-17

**Decision:**

Accept (poster)

**Comment:**

The paper proposes a recommendation algorithm, HPSERec, to address the long-tail problem in sequential recommendation. The approach leverages hierarchical partitioning based on a data imbalance metric, progressive knowledge distillation, and Sinkhorn optimal transport-based feedback.

Most reviewers recognized the following strengths:
* The proposed algorithm is novel, with well-integrated components.
* The method demonstrates consistent improvements over competitive baselines, including recent LLM-based approaches, across multiple datasets and scenarios (tail users/items and overall ranking). Ablation studies further validate the contribution of each module.

Reviewers also noted several weaknesses: the lack of baselines specifically designed for long-tail recommendation, the computational cost of the method, and the absence of modeling for temporal dynamics that are common in real-world recommendation scenarios. Some of these concerns were addressed in the rebuttal. The authors are encouraged to incorporate the additional results presented in the rebuttal, as well as to address the remaining issues where possible, in the final version of the paper.